# LEARNING TO GROUND VLMS WITHOUT FORGETTING

## ABSTRACT

Spatial awareness is key to enable embodied multimodal AI systems. Yet, without vast amounts of spatial supervision, current Visual Language Models (VLMs) struggle at this task. In this paper, we introduce LynX, a framework that equips pretrained VLMs with visual grounding ability without forgetting their existing image and language understanding skills. To this end, we propose a *Dual Mixture of Experts* module that modifies only the decoder layer of the language model, using one frozen Mixture of Experts (MoE) pre-trained on image and language understanding and another learnable MoE for new grounding capabilities. This allows the VLM to retain previously learned knowledge and skills, while acquiring what is missing. To train the model effectively, we generate a high-quality synthetic dataset we call SCouT, which mimics human reasoning in visual grounding. This dataset provides rich supervision signals, describing a step-by-step multimodal reasoning process, thereby simplifying the task of visual grounding. We evaluate LynX on several object detection and visual grounding datasets, demonstrating strong performance in object detection, zero-shot localization and grounded reasoning while maintaining its original image and language understanding capabilities on seven standard benchmark datasets.

## 1 INTRODUCTION

Visual language models (VLMs) have significantly advanced multimodal vision and language tasks, enabling impressive capabilities such as image captioning and visual question answering (Alayrac et al., 2022; Li et al., 2023; Dai et al., 2024; Liu et al., 2024). Models like CLIP (Radford et al., 2021a) leveraged extensive image-caption data for multimodal training, while generative models like Flamingo (Alayrac et al., 2022) and BLIP2 (Li et al., 2023) generate descriptive captions for images. Because of their caption-based nature, these models often lack object localization abilities, making them less suited for applications requiring precise spatial understanding (Wen et al., 2023; Luo et al., 2024; Driess et al., 2023; Jin et al., 2023; Cheng et al., 2024). Naturally, one can equip a model with localization ability by pre-training, (Wang et al., 2023; Chen et al., 2023b). However, this requires massive datasets, human-annotated bounding boxes, and substantial computational resources, making it costly and impractical for smaller setups. Rather than pre-training from scratch, we aim to equip a pre-trained VLM with spatial understanding by fine-tuning.

Closest to our work is PIN (Dorkenwald et al., 2024), which fine-tunes a VLM for the specific task of object localization by adding learned spatial parameters to the vision encoder. Trained on a synthetic dataset of superimposed object renderings, PIN is evaluated on single-object localization, predicting a bounding box given a query object name. Despite the obtained ability for object localization, PIN suffers from catastrophic forgetting, losing image understanding abilities after fine-tuning. Moreover, its synthetic data lacks inter-object relationships, limiting its utility for more complex tasks beyond object localization, like multi-object detection and reasoning (Wang et al., 2023; Chen et al., 2023b). Additionally, as demonstrated by PIN, even parameter-efficient methods like LoRA (Hu et al., 2021) underfit due to the complexity differences between image understanding and grounding tasks. These challenges underscore the need for a solution that adds grounding capabilities for many tasks without compromising a model's pre-existing strengths, which is the focus of our work.

Specifically, we introduce LynX (Linking eXperts for visual grounding), a novel framework that leverages a Dual Mixture of Experts (MoE) architecture. This design allows the model to specialize in both image understanding and visual grounding simultaneously, preventing the catastrophic forgetting seen in PIN and enabling fine-tuning for grounding without sacrificing existing capabilities.

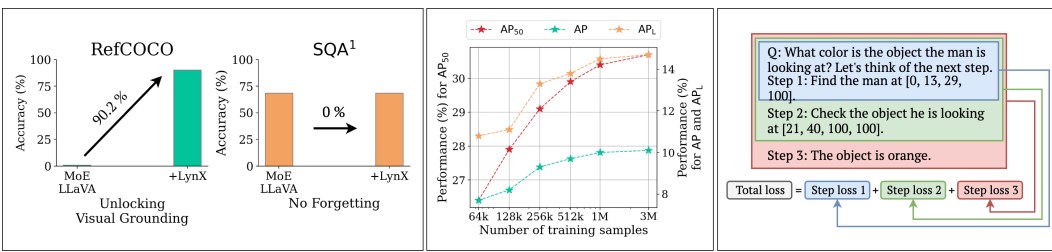

a. New skills + no forgetting.    b. Scaling potential of our SCouT.    c. Our model is trained step-by-step.

Figure 1: **Contributions overview.** Our contributions include **(a)** enabling a pre-trained caption-based vision-language model to learn new grounding skills by fine-tuning without forgetting old ones, **(b)** improving model performance by scaling the generated dataset, and **(c)** enabling visual grounding tasks through step-by-step training on our synthetic dataset.

Further, to address the shortcomings of PIN's localization-only dataset, we propose SCouT: Synthetic Chain-of-Thought with Grounding, a high-quality synthetic dataset with step-by-step grounded chain-of-thought annotations. Unlike PIN's object-pasting approach, SCouT captures meaningful spatial relationships and reasoning steps, providing a richer training signal for grounding tasks. We complement this with a step-by-step training methodology inspired by Lightman et al. (2023), breaking down tasks into intermediate steps with individual loss functions, providing clearer learning signals for handling complex multimodal tasks. Additionally, to address the challenge of evaluating VLMs with free-form grounded responses—where existing metrics fall short—we propose an open-source evaluation metric to fairly compare performance.

Our contributions can be summarized as follows:

1. We introduce LynX, a Dual Mixture of Experts framework that enables VLMs to acquire new grounding capabilities via fine-tuning without forgetting pre-trained skills (Figure 1.a).

2. We present SCouT, a high-quality synthetic dataset with step-by-step grounded chain-of-thought annotations, specifically designed to facilitate effective training of VLMs on grounding tasks (Figure 1.b).

3. We propose a step-by-step training methodology, breaking down tasks into intermediate steps with individual loss functions, improving model performance by scaling the generated dataset (Figure 1.c).

4. We introduce an open-source evaluation pipeline for VLMs on object detection tasks, accommodating their open-ended generative nature.

## 2 RELATED WORK

**Caption-based Visual Language Models (VLMs).** Large language models, efficient at instruction following and generalization, have been seamlessly integrated with vision-only encoder models, yielding impressive results in multimodal tasks (Alayrac et al., 2022; Li et al., 2023; Bai et al., 2023; Chen et al., 2023a; Wang et al., 2023; Chen et al., 2023d; Zhang et al., 2023a; Lin et al., 2023; Cha et al., 2023; Dai et al., 2024; Ye et al., 2023; Zhao et al., 2023; Chen et al., 2023c; Liu et al., 2023; Zhang et al., 2023b). Flamingo (Alayrac et al., 2022) and BLIP-2 (Li et al., 2023) are pioneering works in this area. Flamingo combines a pretrained CLIP (Radford et al., 2021b) image encoder with a pretrained LLM using perceiver and gated cross-attention blocks, while BLIP-2 employs a lightweight Querying Transformer, pretrained in two stages: vision-language representation from a frozen image encoder and vision-to-language generation from a frozen language model. Subsequently, recent works have focused on improving performance through optimizing training strategies (Bai et al., 2023; Chen et al., 2023a), increasing resolution of image (Bai et al., 2023; Chen et al., 2023a; Wang et al., 2023), enhancing image encoders (Chen et al., 2023d; Zhang et al., 2023a; Bai et al., 2023), aligning the input (Lin et al., 2023) and projection layers (Cha et al., 2023; Alayrac et al., 2022; Dai et al., 2024; Ye et al., 2023; Zhao et al., 2023; Chen et al., 2023c). More importantly, many recent works have focused on improving the model performance by expanding the instruction-tuning dataset (Liu et al., 2023; Zhang et al., 2023b; Zhao et al., 2023). Instruction tuning in these models

typically results in image captioning or simple question answering, overlooking spatial reasoning and object localization. So, they excel at generating descriptive text but struggle with tasks requiring precise spatial comprehension or multi-object grounding. In contrast, our work addresses these gaps by enabling spatial understanding and complex visual reasoning, extending beyond captioning to tasks like visual grounding and object localization.

**Grounded Visual Language Models.** Extending the capabilities of VLMs beyond image and language understanding, many models now enable visual grounding to localize objects in an image (Chen et al., 2021; Wang et al., 2022b; Lu et al., 2022; Yang et al., 2022; Wang et al., 2022a; 2024; Chen et al., 2023b; Wang et al., 2023; Bai et al., 2023; Dorkenwald et al., 2024). Pix2Seq (Chen et al., 2021) formulates object detection as an auto-regressive language modeling task, conditioned on observed input pixels. Inspired by this approach, models such as OFA (Wang et al., 2022b), Unified-IO (Lu et al., 2022), UniTab (Yang et al., 2022), GIT (Wang et al., 2022a), and VisionLLM (Wang et al., 2024) incorporate coordinate vocabulary alongside language vocabulary for grounding tasks. In contrast, models like Shikra (Chen et al., 2023b), CogVLM (Wang et al., 2023), and Qwen-VL (Bai et al., 2023) treat positional input/output as natural language, demonstrating the ability to generate interleaved grounded visual captions for images. While these models perform impressively, they need large annotated datasets and significant computational resources. To address this, PIN (Dorkenwald et al., 2024) introduces a learnable positional insert module and a synthetic dataset to enhance spatial understanding in pretrained VLMs. However, their method shows limited grounding performance and leads to forgetting pretrained knowledge. In contrast, we introduce LynX, a framework that equips VLMs with grounding capabilities while retaining their image understanding skills.

## 3 METHOD

In the following sections, we briefly review standard VLMs and the concept of Mixture of Experts (MoE). We then introduce *LynX*, our dual MoE framework that equips VLMs with grounding abilities while preserving their image understanding skills, without extensive pretraining. We also present *SCouT*, a synthetic dataset with step-by-step reasoning supervision for learning complex grounding tasks. Finally, we explain how *step-by-step* learning modifies the training loss to integrate grounding and reasoning within the Dual MoE framework.

### 3.1 PRELIMINARIES

**Visual Language Models (VLMs).** VLMs process both image and text data for multimodal generative tasks. These models consist of a vision encoder $\psi(\cdot)$, a language decoder, $\phi(\cdot)$, and a mapper function $f(\cdot)$. The language decoder takes a sequence of tokens as inputs $[v_1, v_2, \ldots, v_m, t_1, t_2, \ldots, t_n]$ being composed of visual and textual tokens. Visual tokens are computed from an image $\mathbf{x}$ as $[v_1, v_2, \ldots, v_m] = f(\psi(\mathbf{x}))$, and textual tokens are computed from the text input $\mathbf{t}$ as $[t_1, t_2, \ldots, t_n] = $ Tokenizer($\mathbf{t}$). VLMs are trained via cross-entropy loss on a next-token prediction task.

**Mixture of Expert.** Mixture of Experts (MoEs) are a way to increase the small model capacity to compete with large models performance without a proportional increase in computational cost (Shazeer et al., 2017). Specifically, a MoE layer is composed of $E$ "experts" and a gating network $g(\cdot)$. The gating network decides which expert is most suitable for a given token:

$$l_n = \text{MoE}(l_{n-1}) = \sum_{i=1}^{E} g_i(l_{n-1}) \cdot e_i(l_{n-1}), \tag{1}$$

where $l_n$ represents the output of the $n$-th layer, $l_{n-1}$ the input, $E$ the total number of experts, $g_i(\cdot)$ the gating function's weight for the $i$-th expert, and $e_i(\cdot)$ the $i$-th expert's output. During inference, only the top-$k$ experts can be used, reducing inference costs considerably.

### 3.2 LYNX: LINKING EXPERTS FOR VISUAL GROUNDING

We start with a caption-based mixture-of-expert VLM (Lin et al., 2024) adept at visual question answering tasks, and extend it for the task of object grounding as depicted in Figure 2. A transformer

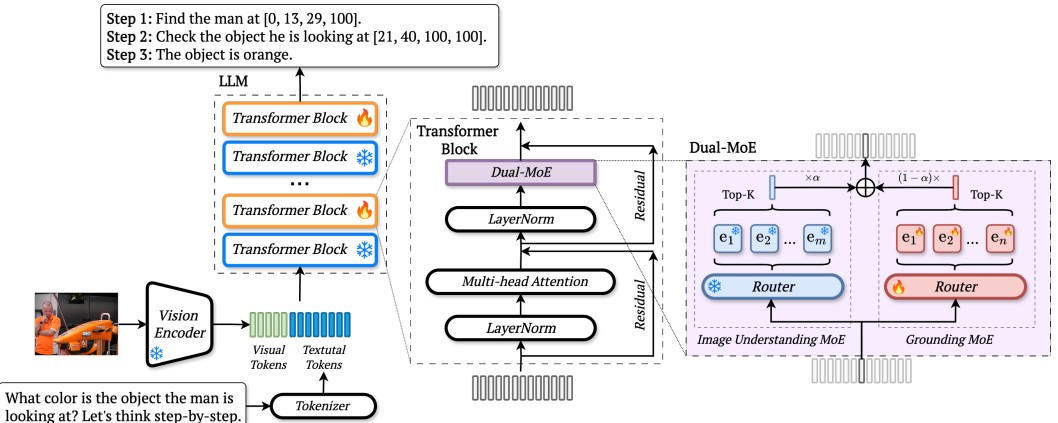

Figure 2: **LynX architecture overview.** Our dual Mixture of Experts (MoE) module replaces the feed-forward network in every other transformer block of the language decoder, preserving the visual encoder and self-attention modules. It includes two parallel MoE blocks: one frozen for image understanding and one trainable for visual grounding. Outputs are combined using a learnable $\alpha$ coefficient, facilitating knowledge transfer and faster training. During inference, alpha is adjusted based on the task, classified by our BERT token classifier.

block of the language decoder of a VLM is composed of multi-head attention (MHA) and feed-forward network (FFN), which processes the input tokens as follows:

$$\hat{l}_n = \text{MHA}(\text{LN}(l_{n-1})) + l_{n-1}, \tag{2}$$

$$l_n = \text{FFN}(\text{LN}(\hat{l}_n)) + \hat{l}_n, \tag{3}$$

where $l_{n-1}$ is the input from layer $n-1$, $\hat{l}_n$ is the hidden representation at layer $n$, and $l_n$ is the output of the $n$-th layer. The mixture of expert module only modifies Eq. (3) by replacing the FFN module with a MoE in the transformer block computation as follows:

$$l_n = \text{MoE}(\text{LN}(\hat{l}_n)) + \hat{l}_n. \tag{4}$$

We introduce a parallel MoE module for visual grounding and modify above equations as follows:

$$l_n^{\text{IU}} = \text{MoE}^{\text{IU}}(\text{LN}(\hat{l}_n)) + \hat{l}_n, \quad l_n^{\text{VG}} = \text{MoE}^{\text{VG}}(\text{LN}(\hat{l}_n)) + \hat{l}_n, \tag{5}$$

$$l_n = \alpha \cdot l_n^{\text{IU}} + (1-\alpha) \cdot l_n^{\text{VG}}, \tag{6}$$

where $\text{MoE}^{\text{IU}}$ is a frozen MoE module pretrained on image understanding tasks, $\text{MoE}^{\text{VG}}$ is a learnable MoE module trained for object localization task, and $\alpha$ is a learnable coefficient weight adjusting the contribution of each MoE module. This design choice prevents catastrophic forgetting of pretrained image understanding skills of VLMs. Moreover, the shared modules allow knowledge transfer from the pretrained MoE into the grounding MoE, helping the latter to better interpret grounding tasks.

**Training step.** We train our method using a cross-entropy loss for the next token prediction task:

$$L = - \left[ \sum_{i=1}^{N} \log P_\theta(t_i | v_1, \ldots, v_m, t_1, \ldots, t_{i-1}) \right] + \lambda \cdot R(g), \tag{7}$$

where $L$ is the next token prediction loss, $N$ represents the length of the text sequence, $v_i$ refers to the $i$-th visual token in the sequence, $t_i$ denotes the $i$-th textual token in the sequence, $\theta$ refers to the model parameters, $\lambda$ is a regularization coefficient, and $R(g)$ is a regularization term for sparsifying the gating mechanism. This loss function aims to minimize the discrepancy between the predicted and actual next token in the sequence.

**Inference step.** During inference, we automatically determine the task type (image understanding or visual grounding) based on the input prompt and adjust $\alpha$ accordingly. Instead of relying on manual task tokens, we employ a lightweight BERT-based classifier (Devlin, 2018) which takes an input

prompt and classifies it into one of the two task categories. Based on the classifier's output, we adjust $\alpha$ dynamically:

$$\alpha = \begin{cases} 1 & \text{for Image Understanding }, \\ \text{unchanged} & \text{for Visual Grounding.} \end{cases} \quad (8)$$

Thus, at test time, the output of the dual MoE module is as follows:

$$l_{n+1} = \begin{cases} l_{n+1}^{\texttt{IU}} & \text{for Image Understanding,} \\ \alpha \cdot l_{n+1}^{\texttt{IU}} + (1-\alpha) \cdot l_{n+1}^{\texttt{VG}} & \text{for Visual Grounding.} \end{cases} \quad (9)$$

This automated task classification eliminates the need for manual task tokens, making the system more user-friendly and robust. The BERT classifier adds minimal computational overhead, as it is an 8-bit quantized tiny model with approximately 1 million parameters, bringing the total active parameters from 1.67B to 1.671B. Our experiments show that the classifier achieves 99.98% accuracy, ensuring negligible impact on performance.

### 3.3 SCOUT: SYNTHETIC CHAIN-OF-THOUGHT WITH GROUNDING

Visual question answering datasets often incorporate spatial reasoning, such as *"What object is to the left of the girl?"* or *"Is there a bowl on top of the table?"*. Grounding tasks particularly benefit from this spatial reasoning, as describing relationships like "A cat at [x1, y1, x2, y2] sits to the left of a dog at [a1, b1, a2, b2]" offers clearer relative positioning, improving the interpretation of localization data for VLMs. Following this intuition, recent works such as Shikra (Chen et al., 2023b) have made progress in creating grounded chain-of-thought datasets. Shikra uses an LLM to generate reasoning-based question-answer pairs from image captions, without access to the actual visual content. However, this reliance on captions alone leads to hallucinated narratives that do not reflect the image (see halluciantion examples of Shikra dataset in Appendix Figure 5).

To address the hallucination problem, we introduce the *SCouT* dataset, which integrates visual information with textual descriptions from captions. We generate "where" and "what" questions using an LLM, such as Mixtral, ensuring contextual alignment with captions through in-context prompting (see Appendix Figure 7). For answer generation, instead of relying solely on the LLM like Shikra, we use a pretrained VLM, such as CogVLM, which incorporates image context to answer questions step-by-step, reducing hallucinations (see Appendix Figure 3). In a comparison of 100 randomly selected samples from each dataset, SCouT achieved an accuracy of 94.7%, significantly outperforming Shikra's 63.1%. A correct response is defined as one where the generated grounding steps accurately reflect the relationships and spatial positions of objects in the image, without hallucinations. This demonstrates the effectiveness of our method in producing reliable, contextually grounded data.

**Step-by-step loss function.** Inspired by Lightman et al. (2023), we then decompose the generated answers, focusing on one task per sentence, to make it easily digestible for training our small visual language model, as seen in Figure 1 (c). These steps are not separate tasks but subtasks of a unified task. To illustrate this concept mathematically, the loss function for training under step-by-step reasoning supervision can be expressed as follows:

$$L_{\text{step-by-step}} = \sum_{j=1}^{J} \left[ -\left( \sum_{i=1}^{N_j} \log P_\theta(t_i^{(j)} \mid v_1, \ldots, v_m, t_1^{(j)}, \ldots, t_{i-1}^{(j)}) \right) \right] + \lambda \cdot R(g), \quad (10)$$

.

where $L_{\text{step-by-step}}$ represents the step-by-step reasoning loss function, $J$ is the number of reasoning steps, $N_j$ is the number of tokens in step $j$, $t_i^{(j)}$ represents the $i^{th}$ token in the $j^{th}$ step output, $v_1 \ldots v_m$ are the image tokens, $P_\theta$ is the probability predicted by the model and $R(g)$ is the regularization term with weight $\lambda$. We compare the dataset statistics of our generated dataset with Shikra's in Table 10 and provide detailed visualizations of our dataset in Figures 3 and 4 in the appendix.

## 4  EXPERIMENTS

We perform our evaluation of LynX on three different grounding tasks: i) object localization, ii) object detection and iii) visual grounding, on top of standard image understanding tasks. The details of our architectural implementation and the dataset splits for each task is explained as follows.

**Implementation details.**  LynX is built on MoE-llava (Lin et al., 2024), which uses Phi2 as its pretrained language model. MoE-llava has 4 experts dedicated to image understanding tasks. For LynX, we add a separate MoE module with two experts for grounding tasks, initialized from the image understanding MoE in the same decoder layer. The vision encoder and multi-head attention layers remain frozen, as do the interleaved decoder layers. LynX is optimized with AdamW (Loshchilov & Hutter, 2019) using a $2e - 5$ learning rate, and trained on $4 \times$ A6000 GPUs for about 1.5 days. LynX contains 1.67B trainable parameters, with only 0.8B active, roughly one fourth the size of Shikra-7B.

**Datasets.**  We train LynX using three types of datasets:

**1. Referential expression comprehension (REC):** The task is to use textual references to accurately identify and locate objects in a scene. For example, given the phrase "guy with his back turned to us," the model must find the person and provide the bounding box coordinates around him in the image (see Figure 6 in the appendix). We use the standard RefCOCO (Yu et al., 2016) dataset which consists of three splits: RefCOCO, RefCOCO+ and RefCOCO$_g$. In total, we have 128,000 samples of image-referential expression pairs from the COCO2014 (Lin et al., 2014) train dataset.

**2. Grounded image captioning (GIC):** The task involves generating detailed image captions that not only identify objects in the image but also provide precise bounding box coordinates for each mentioned object (see Figure 6 in the appendix). For this task, we process $108k$ images from the COCO2017 (Lin et al., 2014) train dataset using CogVLM to produce interleaved captions with corresponding bounding boxes for each object described.

**3. Synthetic grounded visual question answering (SCouT):** This task requires the object to have image understanding and spatial reasoning along with localization capability (see section 3.3 for more detail). We use the Flickr30k (Plummer et al., 2015) image dataset consisting of image captions. We prompt Mixtral (Jiang et al., 2024) to generate questions from these captions by leveraging in-context learning ability of LLMs (Figure 7 in appendix). Given the generated questions and the corresponding image, we prompt CogVLM to generate step-by-step answers for each query. For all tasks, bounding box locations are integers, with $x$ and $y$ coordinates ranging from 0 to 99, creating a $100 \times 100$ grid.

### 4.1  OBJECT LOCALIZATION

We begin by comparing LynX with PIN (Dorkenwald et al., 2024), as both methods involve fine-tuning pretrained models for grounding tasks. PIN evaluates its object localization ability on a task that requires generating bounding boxes when prompted with object names. Their evaluation is conducted on subsets of COCO, PVOC, and LVIS, with up to three objects per image, totaling 3,582, 2,062, and 6,016 test images, respectively. For each image, the model is provided with a ground truth object name and tasked with localizing it. The mean Intersection over Union (mIoU) is reported for all bounding boxes, as well as for medium ($32 \times 32$ to $96 \times 96$ pixels) and large (over $96 \times 96$ pixels) bounding boxes, quantifying the overlap between the predicted and true bounding boxes.

LynX outperforms PIN in single-object localization, with a $22\%$ improvement in mIoU on PVOC, $32\%$ on COCO, and $39\%$ on LVIS, particularly excelling with medium-sized objects. Notably, LynX achieves this without being exclusively trained for localization, whereas PIN, despite being tailored for this specific task, struggles to match our performance. Attempts to fine-tune MoE-LLaVA using LoRA also underperform across all datasets, reinforcing the robustness of our dual MoE architecture for both localization and image understanding.

Although PIN uses fewer parameters (1.4M vs. 1.67B), it is limited to single-object detection and suffers from catastrophic forgetting of its image understanding capabilities. This disparity in backbones may introduce unfairness in the comparison. To better showcase LynX's full potential, we introduce a protocol-based evaluation that assesses more complex multi-object grounding tasks – areas where PIN cannot compete – and standardizes evaluation for comparing VLMs.

Table 1: **Comparison with PIN** (Dorkenwald et al., 2024) on their COCO, PVOC, and LVIS subsets. Our model outperforms PIN across all datasets and metrics, despite not being specifically fine-tuned for this task.

| Method | PVOC$_{\leq 3 \text{ Objects}}$ | | | COCO$_{\leq 3 \text{ Objects}}$ | | | LVIS$_{\leq 3 \text{ Objects}}$ | | |
|---|---|---|---|---|---|---|---|---|---|
| | mIoU | mIoU$_M$ | mIoU$_L$ | mIoU | mIoU$_M$ | mIoU$_L$ | mIoU | mIoU$_M$ | mIoU$_L$ |
| PIN | 0.45 | 0.27 | 0.62 | 0.35 | 0.26 | 0.59 | 0.26 | 0.24 | 0.61 |
| MoE-LLaVA w/ LoRA | 0.43 | 0.21 | 0.65 | 0.36 | 0.29 | 0.60 | 0.24 | 0.21 | 0.62 |
| LynX | **0.68** | **0.58** | **0.81** | **0.66** | **0.57** | **0.78** | **0.65** | **0.55** | **0.76** |

## 4.2 OBJECT DETECTION

While the comparison with PIN highlights LynX's superiority in standard localization tasks, we aim to demonstrate its full potential in more complex visual grounding challenges. Unlike PIN, which is limited to single-object detection, LynX handles multi-object localization and complex grounding across multiple entities. However, traditional object detection datasets like COCO (Lin et al., 2014), PASCAL VOC (Everingham et al., 2010), and LVIS (Gupta et al., 2019) are not fully suited for VLMs, as LynX generates free-form captions and predicts objects from an open vocabulary. To fairly evaluate its performance against larger models like CogVLM and Shikra we introduce a protocol-based framework that bridges this gap.

**Protocol 1 - Common Class Comparison:** In this evaluation, CogVLM serves as the ground truth, providing object names and bounding boxes. The task asks models to "Locate objects in this image along with their bounding box coordinates," with responses in free-form text. We focus on the top 50 shared object classes between each model and CogVLM. LynX—after incorporating SCouT into its training—significantly outperforms all models, including Shikra 7B, with an increase of AP$_{50}$ by 7.4 on COCO, 6.8 on PVOC, and 7.9 on LVIS. Despite having fewer parameters and no pretraining, LynX surpasses Shikra in these metrics, while PIN scores zero as it only generates bounding boxes without object names. The importance of dataset quality is evident in LynX's performance gains. Initial training with the REC dataset resulted in low scores, as single-object annotations failed to generalize to multi-object detection. Adding the GIC dataset led to improvements of 28.5 on COCO, 31.7 on PVOC, and 27.6 on LVIS. However, incorporating Visual Genome (VG)—a noisy dataset—negatively impacted performance, highlighting the detrimental effect of noisy data. Replacing VG with SCouT yielded substantial improvements, demonstrating the necessity of high-quality, contextually grounded annotations for complex tasks.

**Protocol 2 - Class Grouping:** In this protocol, we map open-vocabulary predictions from LynX and CogVLM to predefined categories in COCO, PVOC, and LVIS using a sentence transformer. This groups similar classes (e.g., "man" and "woman" into "person"), standardizing labels across models. LynX—trained with SCouT continues to outperform other models under these stricter class mappings, with AP$_{50}$ increasing by 5.7 on COCO, 6.0 on PVOC, and 3.9 on LVIS compared to the GIC variant. As in Protocol 1, SCouT proves essential for maintaining strong performance, while the noise in VG continues to hinder results.

**Protocol 3 - Reference Ground Truth Annotation:** In this protocol, we use COCO's standardized annotations as the ground truth to ensure a fair comparison between VLMs, as relying on model-generated outputs (as in Protocols 1 and 2) can be misleading due to different object focuses by different VLMs. By using COCO annotations, we objectively evaluate each model's accuracy in detecting objects. LynX, despite being much smaller (1.67B parameters), performs comparably to CogVLM (17B), our dataset generator and upper bound. LynX also outperforms Shikra (7B), even though both Shikra and CogVLM have significantly more parameters and pretraining. The relatively low average

Table 2: **Comparison on COCO for Protocol 3.** LynX peform comparably to CogVLM and surpasses Shikra7B, despite fewer parameters. Low scores highlight the challenge of this task, where VLMs predict less objects than dataset annotations.

| Method | AP↑ | AP$_{50}$↑ | AP$_L$↑ |
|---|---|---|---|
| PIN | 0 | 0 | 0 |
| Shikra | 13.2 | 46.8 | 16.7 |
| LynX | 14.0 | 48.3 | 17.3 |
| CogVLM | 16.1 | 52.7 | 21.3 |

precision (AP) scores across all models highlight the difficulty of this task—COCO averages 7 annotations per image, while VLMs typically predict 3-4 objects. This protocol provides a standardized

way to evaluate object detection across VLMs, demonstrating LynX's strong balance of model size and performance.

Table 3: **Results on the COCO, PVOC, and LVIS validation sets for Protocols 1 and 2.** LynX, trained with SCouT, consistently outperforms other variants across all evaluated metrics. In Protocol 1, despite being a zero-shot evaluation with training only on COCO, LynX generalizes effectively to PVOC and LVIS without prior exposure. In Protocol 2, which maps open-vocabulary predictions to predefined object categories, LynX maintains strong performance, demonstrating robustness even under class-mapping constraints. Adding the Visual Genome (VG) dataset negatively impacts performance, emphasizing the need for high-quality, contextually grounded data like SCouT for effective training on complex grounding tasks.

| Dataset Type | | | | COCO | | | VOC | | | LVIS | | |
|---|---|---|---|---|---|---|---|---|---|---|---|---|
| REC | GIC | VG | SCouT | AP↑ | AP$_{50}$↑ | AP$_L$↑ | AP↑ | AP$_{50}$↑ | AP$_L$↑ | AP↑ | AP$_{50}$↑ | AP$_L$↑ |
| **Protocol 1: Common Class** | | | | **50 Classes** | | | **50 Classes** | | | **50 Classes** | | |
| ✓ | ✗ | ✗ | ✗ | 0 | 0 | 0 | 0 | 0 | 0 | 0 | 0 | 0 |
| ✓ | ✓ | ✗ | ✗ | 8.2 | 28.5 | 10.1 | 10.3 | 31.7 | 12.4 | 8.0 | 27.6 | 9.6 |
| ✓ | ✓ | ✓ | ✗ | 6.5 | 23.1 | 8.2 | 8.6 | 29.7 | 11.0 | 6.2 | 24.1 | 7.9 |
| ✓ | ✓ | ✓ | ✓ | 10.4 | 34.7 | 13.0 | 12.5 | 37.2 | 15.1 | 10.3 | 34.2 | 13.5 |
| ✓ | ✓ | ✗ | ✓ | 11.1 | 35.9 | 13.7 | 13.1 | 38.5 | 15.9 | 11.1 | 35.5 | 14.0 |
| **PIN** | - | - | - | 0 | 0 | 0 | 0 | 0 | 0 | 0 | 0 | 0 |
| **Shikra 7B** | - | - | - | 8.9 | 29.3 | 12.7 | 11.7 | 33.8 | 13.6 | 9.2 | 28.5 | 10.2 |
| **Protocol 2: Class Grouping** | | | | **80 Classes** | | | **20 Classes** | | | **200 Classes** | | |
| ✓ | ✗ | ✗ | ✗ | 0 | 0 | 0 | 0 | 0 | 0 | 0 | 0 | 0 |
| ✓ | ✓ | ✗ | ✗ | 6.9 | 23.4 | 9.4 | 12.1 | 38.0 | 15.5 | 4.1 | 12.1 | 5.5 |
| ✓ | ✓ | ✓ | ✗ | 4.8 | 20.3 | 7.2 | 10.1 | 36.5 | 13.2 | 2.4 | 9.7 | 4.0 |
| ✓ | ✓ | ✓ | ✓ | 8.7 | 27.8 | 12.2 | 13.7 | 41.0 | 17.1 | 5.2 | 14.8 | 6.9 |
| ✓ | ✓ | ✗ | ✓ | 9.3 | 29.1 | 13.3 | 14.5 | 44.0 | 19.2 | 5.8 | 16.0 | 7.8 |
| **PIN** | - | - | - | 0 | 0 | 0 | 0 | 0 | 0 | 0 | 0 | 0 |
| **Shikra 7B** | - | - | - | 7.2 | 25.5 | 10.8 | 13.3 | 39.1 | 16.7 | 4.9 | 13.4 | 5.8 |

## 4.3 VISUAL GROUNDING

We evaluate LynX on two key visual grounding tasks: referential expression comprehension (REC) and phrase grounding. For REC, we use the RefCOCO, RefCOCO+, and RefCOCOg (Yu et al., 2016) datasets, where the objective is to identify a single object in an image based on a descriptive query. In contrast, phrase grounding, evaluated on the Flickr30k Entities (Plummer et al., 2015) dataset, involves linking multiple objects to their corresponding noun phrases in a sentence, requiring more complex contextual reasoning.

Unlike most models in this comparison, which rely on pretraining, LynX and PIN are fine-tuning approaches. Despite this, LynX trained with SCouT outperforms Shikra-7B on the complex Flickr30k phrase grounding task by 1.4%, demonstrating our model's strength in relational tasks. On REC tasks, LynX remains highly competitive despite Shikra's specialized training, particularly given our smaller model size and fine-tuning approach. We also outperform OFA-L and VisionLLM-H on all REC benchmarks and exceed PIN by 62% on RefCOCO test-A, where PIN struggles with more complex queries. The SCouT variant further improves performance, highlighting the value of step-by-step supervision.

## 4.4 IMAGE UNDERSTANDING EVALUATION

A major strength of LynX is its ability to retain image understanding capabilities even after fine-tuning for grounding tasks—unlike PIN, which suffers from catastrophic forgetting. As shown in Table 5, LynX matches the performance of MoE-LLaVA and is even better than much larger models like LLaVA-phi2 (Liu et al., 2024), despite being nearly ten times smaller. This is achieved without the extensive pretraining on multiple datasets that models like Shikra require, which cannot even perform

Table 4: **Comparison of LynX variants on the REC task.** We compare LynX with OFA-L, VisionLLM-H, Shikra, and PIN. LynX outperforms OFA-L, VisionLLM-H, and PIN, with the SCouT-trained variant showing enhanced performance compared to the variant trained without it. We also report results on the Flickr30k Entities dataset for both Shikra-7B and LynX. Numbers for OFA-L, VisionLLM-H, and Shikra are taken from Shikra (Chen et al., 2023b).

| Model | RefCOCO | | | RefCOCO+ | | | RefCOCOg | | Flickr30k Entities | |
|---|---|---|---|---|---|---|---|---|---|---|
| | val | test-A | test-B | val | test-A | test-B | val | test | val | test |
| OFA-L* (Wang et al., 2022b) | 80.0 | 83.7 | 76.4 | 68.3 | 76.0 | 61.8 | 67.6 | 67.6 | - | - |
| VisionLLM-H (Wang et al., 2024) | - | 86.7 | - | - | - | - | - | - | - | - |
| Shikra-7B (Chen et al., 2023b) | 87.0 | 90.6 | 80.2 | 81.6 | 87.4 | 72.1 | 82.3 | 82.2 | 75.84 | 76.54 |
| PIN (Dorkenwald et al., 2024) | - | 26.4 | - | - | - | - | - | - | - | - |
| LynX w/ REC | 83.9 | 89.1 | 77.6 | 77.0 | 84.3 | 68.1 | 79.3 | 78.3 | - | - |
| LynX w/ REC + SCouT | 85.4 | 90.2 | 79.3 | 78.4 | 85.7 | 68.5 | 80.3 | 80.6 | 76.83 | 77.91 |

Table 5: **Benchmark evaluation results for LynX on standard image understanding tasks.** Despite its smaller size, LynX performs better than larger models like LLaVA-phi2 and I-80B, successfully retaining the image understanding abilities of its base (MoE-LLava) through the dual MoE module.

| Model | | Image Question Answering | | | Benchmark Toolkit | | | |
|---|---|---|---|---|---|---|---|---|
| | Active | GQA | SQA[1] | VQA$^T$ | POPE | MME | LLaVA$^W$ | MM-Vet |
| I-80B (Laurençon et al., 2024) | 65B | 45.2 | - | 30.9 | - | - | - | - |
| LLaVA-phi2 | 13B | - | 68.4 | 48.6 | 85.0 | 1335.1 | - | 28.9 |
| MoE-LLaVA-phi2 (our base) | 3.6B | 61.4 | 68.5 | 51.4 | 86.3 | 1423.0 | 94.1 | 34.3 |
| LynX | 1.6B | 61.4 | 68.5 | 51.4 | 86.3 | 1423.0 | 94.1 | 34.3 |

these image understanding tasks. The reported numbers, except for MME, reflect accuracy scores, while MME represents a cumulative perception score with a maximum value of 2000.

# 5 ABLATIONS

## 5.1 FINE-TUNING CHALLENGES

From Tables 6 and 7, fine-tuning MoE-LLaVA on both image understanding (task A) and grounding (task B) significantly degrades task A performance, while fine-tuning solely on task B leads to catastrophic forgetting, where task A abilities are entirely lost. Despite substantial training, these methods fall short compared to our Dual MoE approach.

The zero results in Table 6 illustrate catastrophic forgetting: the model generates bounding boxes for task A prompts (e.g., GQA expects "brown" as an answer but receives a bounding box), resulting in zero accuracy. Similarly, the zero average precision (AP) scores in Table 7 reflect the model's inability to generate bounding boxes without specific training, leading to zero AP on MS-COCO. This highlights the necessity of our Dual MoE module, for overcoming the limitations of traditional fine-tuning.

Table 6: **Grounding results on COCO validation set.** LynX achieves better grounding performance compared to MoE-LLaVA finetuned variants.

| Method | AP↑ | AP$_{50}$↑ | AP$_L$↑ |
|---|---|---|---|
| MoE-LLaVA (base) | 0 | 0 | 0 |
| MoE-LLaVA (A & B) | 8.1 | 32.6 | 10.3 |
| MoE-LLaVA (only B) | 10.7 | 35.2 | 12.9 |
| LynX | 11.1 | 35.9 | 13.7 |

## 5.2 EFFECT OF TRAINING LYNX WITH NEGATIVE SUPERVISION DATA.

In constructing the SCouT dataset, we incorporate negative supervision samples (Figure 4 in the appendix) to train LynX. This approach offers two key advantages: First, as demonstrated in Table 8, training with negative samples leads to notable improvements in object detection performance, enhancing the model's ability to distinguish relevant objects. Second, negative supervision significantly reduces hallucinations, a common issue in VLMs. For example, as illustrated in Figure 6

Table 7: **Image understanding results.** Fine-tuning MoE-LLaVA on both task A and B leads to poor performance on image understanding tasks, while fine-tuning only on task B causes catastrophic forgetting of task A. LynX preserves performance across both tasks.

| | GQA↑ | SQA↑ | VQA$^T$↑ | POPE↑ | MME↑ | LLaVA$^W$↑ | MM-Vet↑ |
|---|---|---|---|---|---|---|---|
| MoE-LLaVA (base) | 61.4 | 68.5 | 51.4 | 86.3 | 1423.0 | 94.1 | 34.3 |
| MoE-LLaVA (A & B) | 53.1 | 56.9 | 46.3 | 65.7 | 1347.0 | 71.8 | 28.6 |
| MoE-LLaVA (only B) | 0 | 0 | 0 | 0 | 0 | 0 | 0 |
| LynX | 61.4 | 68.5 | 51.4 | 86.3 | 1423.0 | 94.1 | 34.3 |

(appendix), when faced with a query like "What is the dog doing near the shoreline?" LynX, trained with negative samples, first verifies the presence of the dog before attempting to answer. If no dog is present, and only a girl is in the image, LynX recognizes the question as invalid, thereby avoiding incorrect assumptions and improving overall accuracy.

## 5.3 EFFECT OF NUMBER OF EXPERTS.

We investigate the impact of varying the number of experts in the grounding MoE module, experimenting with configurations that use 2 and 4 experts, corresponding to approximately 1.67B and 3.3B trainable parameters, respectively. While increasing the number of experts to 4 results in a slight performance improvement, the gains are marginal compared to the increased computational cost. This diminishing return suggests that beyond a certain point, additional experts do not significantly enhance model performance for the grounding tasks. As a result, we adopt the 2-expert configuration in all our experiments, striking an optimal balance between performance and efficiency, as reflected in the results across different datasets.

Table 8: **LynX trained with negative samples** shows enhanced object detection performance.

| Pos. | Neg. | AP↑ | AP$_{50}$↑ |
|---|---|---|---|
| ✓ | × | 10.8 | 34.9 |
| ✓ | ✓ | 11.1 | 35.9 |

Table 9: **Effect of number of experts.** 2 experts perform on par with 4, using half the trainable parameters.

| | | RefCOCO | | RefCOCO+ | |
|---|---|---|---|---|---|
| Experts | Param. | test-A | test-B | test-A | test-B |
| 2 | 1.6B | 90.2 | 79.3 | 85.7 | 68.5 |
| 4 | 3.3B | 90.9 | 79.8 | 86.3 | 68.8 |

## 5.4 SCALING PROPERTIES OF OUR SYNTHETIC DATASET.

We explore the scalability of SCouT in Figure 1 (part b). Leveraging our efficient data generation pipeline, we experiment with varying dataset sizes ranging from 64k to 3M samples. As seen in our results, model performance improves consistently with increase in dataset size, particularly up to 512k data points, after which the improvements begin to plateau. This trend highlights the effectiveness of SCouT in providing high-quality, diverse training samples, outperforming conventional human-annotated datasets in scaling the model's capabilities for complex grounding tasks.

**Limitations.** We generated our SCouT dataset using CogVLM and trained LynX with this data, making us dependent on its quality. Since the data quality relies on CogVLM's reasoning capability, this sets an upper bound on our performance.

## 6 CONCLUSION

We introduce LynX, a versatile framework with a dual mixture of experts module, enabling a small VLM to retain pretrained knowledge while acquiring new skills. Training step-by-step with our scalable SCouT dataset provides an effective learning signal and highlights the potential of synthetic data. We also present an open-source evaluation pipeline for comparing VLMs at various granularities. LynX demonstrates SOTA effectiveness in object localization on multiple visual grounding benchmarks and matches pretrained performance on image understanding benchmarks.

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

# A   APPENDIX

The supplementary material consists of the following sections: A.1 Dataset Visualization, A.2 Sample Outputs from LynX, A.3 Number of Parameters and Datasets and A.4 In-context Prompts for Mixtral.

## A.1   DATASET VISUALIZATION AND STATISTICS

We present dataset statistics comparison between our generated and shikra generated grounded chain-of-though datasets in Table 10 along with three visualizations in this subsection: the positive samples of our synthetic grounded chain-of-thought dataset in Figure 3, its negative samples in Figure 4, and samples from the noisy dataset generated by Shikra using LLMs in Figure 5.

**Dataset Statistics** We present a comparison of dataset statistics between our Synthetic Grounded Chain-of-Thought (SCouT) dataset and the Shikra-generated dataset in Table 10. The table highlights key metrics such as the number of images, words, turns, objects, and Q/A pairs. This comparison demonstrates the scale and richness of our SCouT dataset.

Table 10: **Comparison of dataset statistics** between our synthetic data and the Shikra-generated dataset.

|  | **Images** | **Words** | **Turns** | **Objects** | **Q/A Pairs** |
|---|---|---|---|---|---|
| Shikra | 883 | 7106 | 1 | 23692 | 5922 |
| SCouT | 30000 | 15524 | $\sim 4$ | 654314 | 3113763 |

**SCouT: Synthetic Grounded Chain-of-Thought Dataset** This dataset is designed to provide step-by-step answers to questions, thereby simplifying the learning process for our models. For example, in the first row of Figure 3, when asked "What is the color of the hat the man is wearing?", instead of directly trying to answer the question, the dataset breaks down the task into manageable steps:

1. Identify the man in the image.
2. Find the hat he is wearing.
3. Determine the color of the hat.
4. Provide the final answer: "The hat is orange."

This structured breakdown helps our smaller models learn more effectively and quickly by reducing the complexity of the task.

**Negative Samples** In Figure 4, we include some negative samples for our SCouT dataset, where the question is intentionally incorrect or irrelevant to the image. This is done to mitigate the hallucinations of LynX. For instance, in the second row of Figure 4, the image shows a girl at the shoreline, but the question asks, "What is the cat doing near the shoreline?" Our methodology begins by attempting to identify the main object (in this case, the cat). If the model cannot find a cat in the image, it correctly identifies the question as invalid. This type of negative supervision is crucial for training our model to recognize and handle invalid or contradictory queries, thereby improving its robustness and accuracy.

We generate $40k$ negative samples for our dataset along with the $3M$ positive samples. The whole dataset will be released.

**Grounded chain-of-thought dataset by Shikra** Finally, we visualize samples from the dataset generated by Shikra using LLMs in Figure 5. These examples highlight common errors due to the absence of visual context during data generation. For example, in the first row, the question asks "Is the man [260.0, 4.04, 443.0, 349.056] smiling for the picture?" and the ground truth response for this in their dataset is "The image quality doesn't provide enough details to determine if the man [260.0, 4.04, 443.0, 349.056] is smiling or not. Hence, it cannot be confidently answered." We can clearly see from the image that the man is smiling. However, in the Flickr30K dataset, where this image and its captions originate, the captions do not mention that the man is smiling. Shikra uses an LLM to generate data based solely on these captions, without analyzing the image itself. As a result, the LLM states that it cannot determine if the man is smiling because the captions do not provide this

| Image | Question | Answer |
|---|---|---|
|  | What is the color of the hat the man is wearing? | Step 1: Find the man [[10,9,79,100]].
Step 2: Check the hat that it wearing, got the hat [[30,9,79,52]].
Step 3: The hat [[30,9,79,52]] is orange.
Step 4: So the answer is orange. |
|  | What is the young girl sitting on? | Step 1: Find the young girl [[0,18,81,100]].
Step 2: Check the object that it sitting on, got the object [[0,7,100,100]].
Step 3: The girl [[0,18,81,100]] is sitting on the chair.
Step 4: So the answer is chair [[0,7,100,100]].' |
|  | Where does the child jump into? | Step 1: Find the child [[60,32,72,52]].
Step 2: Check the object that it jumping into, got the object [[0,81,100,100]].
Step 3: The child [[60,32,72,52]] jumps into the pool.
Step 4: So the answer is swimming pool [[0,81,100,100]]. |
|  | What is the man playing with on the beach? | Step 1: Find the beach [[0,40,100,70]].
Step 2: Check the man playing on it, got the man [[25,6,51,73]].
Step 3: Check the object that the man is playing with, got the objects [[41,28,54,72]], [[51,9,61,19]].
Step 4: The man [[25,6,51,73]] is playing with a dog [[41,28,54,72]] with a stick [[51,9,61,19]].
Step 5: So the answer is dog [[41,28,54,72]] and stick [[51,9,61,19]]. |
|  | What is the man holding in his right hand? | Step 1: Find the man [[33,29,77,100]].
Step 2: Check the right hand of the man, got the hand [[46,65,56,72]].
Step 3: Check the object that it is holding, got the object [[54,61,72,74]].
Step 4: The man [[33,29,77,100]] is holding the glasses.[[54,61,72,74]]
Step 5: So the answer is glasses. |
|  | What color is the shirt the boy is wearing? | Step 1: Find the boy [[83,60,90,86]].
Step 2: Check the shirt that it wearing, got the shirt [[83,65,90,74]].
Step 3: The shirt [[83,65,90,74]] is blue.
Step 4: So the answer is blue. |
|  | What is the person working on the train wearing? | Step 1: Find the train [[0,0,74,81]].
Step 2: Check the person working on it, got the person [[51,44,71,91]].
Step 3: Check the object that he is wearing, got the object [[55,53,69,79]].
Step 4: The man [[51,44,71,91]] is wearing a vest [[55,53,69,79]].
Step 5: So the answer is vest [[55,53,69,79]]. |

Figure 3: **Visualization of the Grounded Chain-of-Thought Dataset.** Here we provide step-by-step answers to questions, simplifying the learning process. For instance, identifying the man, finding his hat, determining its color, and finally answering that the hat is orange. This structured approach aids in faster and more effective learning for smaller models.

information. Similarly, in the last row, the question asks, "Can you see the girl's [89.0, 4.125, 403.0, 375.0] eyes in the image?" with the ground truth answer stating, "The provided information does not mention the girl's [89.0, 4.125, 403.0, 375.0] eyes, so it's difficult to confidently say if they are visible in the image." However, the girl's eyes are clearly visible in the image. Such errors demonstrate the limitations of generating datasets without access to visual information and underscore the importance of creating a high-quality dataset like our SCouT.

| Image | Question | Answer |
|---|---|---|
| | What is the dog doing in the kitchen? | Step 1: Find the dog.
Step 2: There is a woman [[15,22,68,74]] in an olive-brown sweater [[16,35,72,67]] in the image, not a dog. This question is invalid. |
| | What is the cat doing near the shoreline? | Step 1: Find the cat.
Step 2: There is a little girl [[66,6,90,90]] playing along the shoreline and a seagull [[13,10,31,28]] in the image, not a dog. This question is invalid. |
| | Who is holding the pink ball? | Step 1; Find the pink ball.
Step 2: There are two young men [[30,12,99,99]], [[1,2,53,100]] in swimsuits [[49,75,89,100]], [[10,71,47,100]] in the image, not a ball. This question is invalid. |

Figure 4: **Visualization of Negative Samples.** Here we include examples where the question is incorrect or irrelevant, such as asking "What is the cat doing near the shoreline?" when no cat is present. The model begins by identifying the main object and, if it cannot find the object, declares the question invalid. This negative supervision enhances the model's ability to handle invalid or contradictory queries, improving robustness and accuracy.

## A.2 SAMPLE OUTPUTS FROM LYNX

In Figure 6, we showcase outputs generated by our LynX model trained on SCouT. The image demonstrates LynX's versatility across a wide range of tasks, including visual question answering, referential expression comprehension, referential expression grounding, grounded image captioning, and grounded chain of thought. Additionally, LynX effectively avoids hallucination through its chain-of-thought reasoning.

## A.3 NUMBER OF PARAMETERS AND DATASETS

In table 11, we provide the number of trainable parameters of each baseline and the training dataset used for each baseline model used for our paper. As seen, compared the our baselines, LynX efficiently achieves competitive performance with only 1.67 billion trainable parameters and 0.8 billion active parameters, which is significantly less than most models with similar capabilities. Although we have more parameters than PIN, it is limited to generating single bounding box locations per prompt and cannot perform other tasks. Moreover, LynX accomplishes this feat while utilizing a relatively modest training dataset of 651k image-caption pairs, showcasing its ability to extract maximum value from limited data and potentially offering improved scalability and resource efficiency compared to methods requiring billions of parameters or massive training datasets.

## A.4 IN-CONTEXT PROMPTS FOR MIXTRAL

Large Language Models (LLMs) excel in in-context learning scenarios, where they can understand and perform tasks based on provided examples within the input context. This ability allows LLMs to adapt to various tasks without requiring explicit retraining. By leveraging patterns and information from the input context, LLMs can generate coherent and relevant responses, making them highly versatile and effective across diverse applications. Leveraging this quality, we employ Mixtral, an open-source LLM, to generate queries, a crucial step in creating our SCouT dataset. Additionally, we

| Image | Question | Answer |
|---|---|---|
| | Is the man [260.0, 4.04, 443.0, 349.056] smiling for the picture? | The image quality doesn't provide enough details to determine if the man [260.0, 4.04, 443.0, 349.056] is smiling or not. Hence, it cannot be confidently answered. |
| | Can you see any tools in the image? | There is no mention of any tools in the provided context, and based on the image content shared, there seem to be no visible tools. So, the answer is no. |
| | What is the color of the bench [[1.0, 175.875, 499.0, 375.0], [1.0, 123.0, 499.0, 375.0]] the woman [44.0, 39.0, 288.0, 375.0] is sitting on? | The color of the bench [[1.0, 175.875, 499.0, 375.0], [1.0, 123.0, 499.0, 375.0]] the woman [44.0, 39.0, 288.0, 375.0] is sitting on cannot be determined from the image. |
| | Are they wearing any specific type of clothing? | The context provided does not mention any specific type of clothing that the three women [[2.142, 118.0, 102.10199999999999, 421.0], [128.16299999999998, 119.0, 243.11700000000002, 494.0], [242.04600000000002, 104.0, 342.006, 473.0]] are wearing. It only describes their physical appearance as tall and carrying purses on their shoulders. |
| | Is this event happening outdoors? | There is no clear evidence in the image to determine if the event is happening outdoors or indoors. Therefore, I cannot confidently answer this question. |
| | Can you see the girl's [89.0, 4.125, 403.0, 375.0] eyes in the image? | The provided information does not mention the girl's [89.0, 4.125, 403.0, 375.0] eyes, so it's difficult to confidently say if they are visible in the image. |

Figure 5: **Visualization of errors in the Grounded Chain-of-Thought data generated by Shikra** due to absence of visual context. In the first row, the LLM fails to determine if the man is smiling, and in the last row, it cannot confirm the visibility of the girl's eyes, despite both being clearly visible in the images. These errors highlight the limitations of relying solely on captions for data generation.

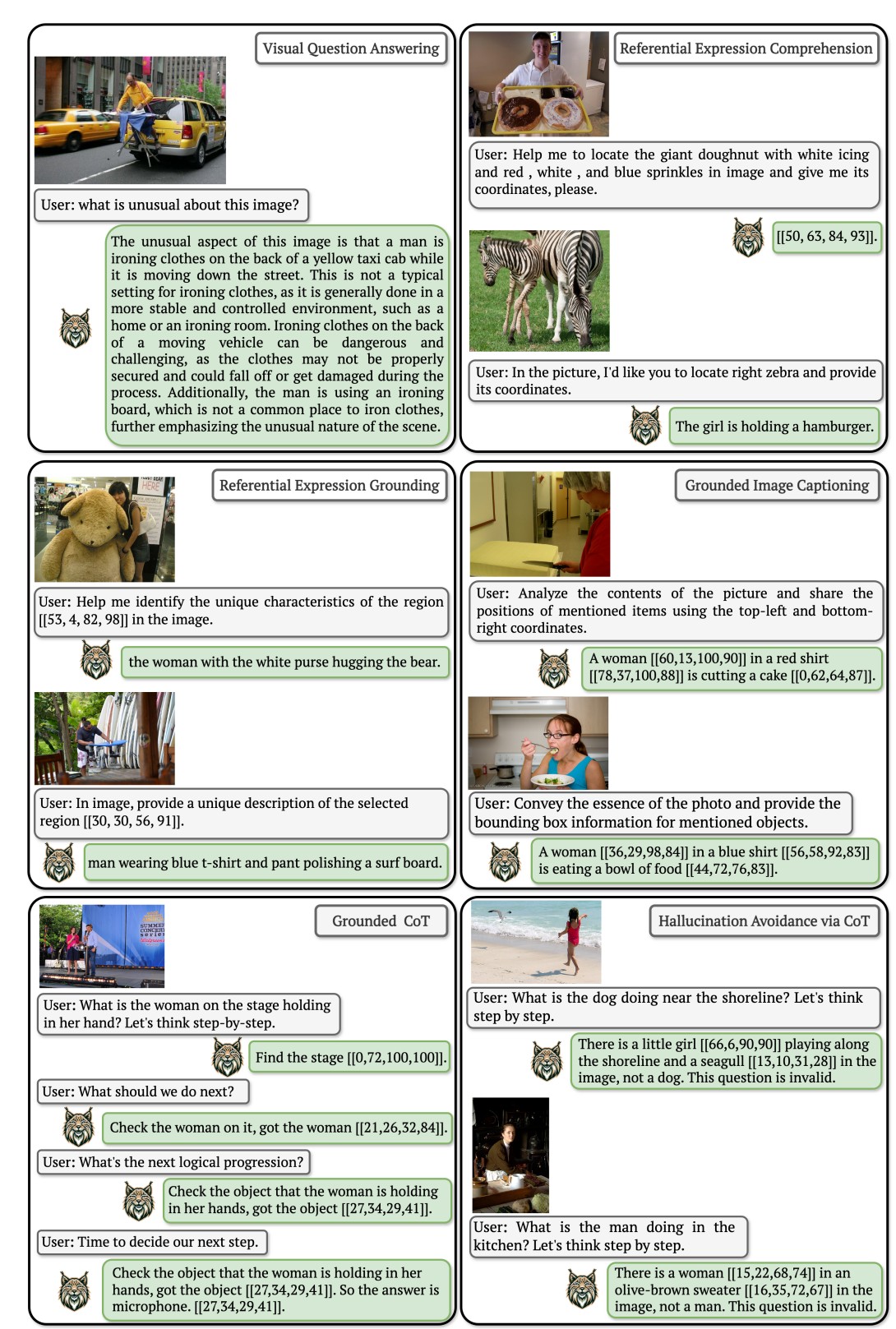

Figure 6: **Samples generated by LynX trained on our SCouT.**

Table 11: Comparison of image-caption models: trainable parameters, active parameters, and training dataset sizes. LynX achieves competitive performance with fewer parameters and a smaller dataset compared to most models with similar capabilities.

| Method | Trainable Parameters | Active Parameters | Size of Training Dataset |
|---|---|---|---|
| PIN | 1.4M | 1.4M | 70k image-caption pairs |
| Shikra | 7B (13B) | 7B (13B) | 7.8M image-caption pairs |
| CogVLM | 17B | 17B | 1B image-caption pairs |
| OFA-L* | 470M | 470M | 24.37M image-caption pairs |
| VisionLLM-H | 1.62B | 1.62B | 738k image-caption pairs |
| I-80B | 80B | 80B | 300M image-caption pairs |
| LLaVA-1.5 | 13B | 13B | 7.8M image-caption pairs |
| LynX | 1.67B | 800M | 651k image-caption pairs |

---

### In-context Positive Grounded Chain of Thought Generation Prompt

[Task Description]: You are an intelligent query generator. I will provide a description of an image with different objects along with their location in an interleaved manner in the [x1, y1, x2, y2] format. Your task is to generate "what", "who", and "where" initiated queries about the objects in the description and how they interact with each other. Limit the answer part of each query to a maximum of three words. Please refer to the example below for the desired format.

[Description]: A man [147, 145, 238, 333] wearing blue overalls [143, 162, 235, 323] is standing next to a red minivan [107, 129, 496, 350] with an open door [195, 148, 298, 299].
Query 1: what is the man wearing the blue overall standing next to? [Answer]: minivan
Query 2: What color is the overall the man is wearing? [Answer]: blue
Query 3: what is the object next to the red minivan? [Answer]: man
Query 4: what is the color of the minivan? [Answer]: red.

[Description]: Two brown dogs [[41, 153, 163, 268], [154, 116, 496, 258]], wearing red collars [[73, 157, 95, 200], [210, 129, 233, 179]] look at each other while running along a dirt field [2, 220, 498, 331].
Query 1: what are the two brown dogs wearing? [Answer]: red collars.
Query 2: What color are the collars of the dogs who are running along the dirt field? [Answer]: red
Query 3: who are wearing the red collars? [Answer]: two dogs.
Query 4: what are the two dogs running on? [Answer]: dirt field.

[Description]: <PLACE_HOLDER>

Figure 7: **Prompt used for generating engaging and relevant questions for the positive samples in our SCouT dataset**, demonstrating Mixtral's ability to enhance query formulation.

utilize this LLM's ability to extract object names and bounding boxes from the free-form text outputs of our VLM models, particularly in grounded image captioning. Figure 7 illustrates the prompt used for generating interesting questions for the positive samples in our SCouT dataset. Figure 8 shows the prompts used to generate negative samples. Finally, Figure 9 depicts the prompts employed to extract objects from grounded image captions produced by our models.

---

### In-context Negative Grounded Chain of Thought Generation Prompt

[Task Description]: You are an intelligent query generator and query solver. I will provide a description of an image with different objects along with their location in an interleaved manner in the [x1, y1, x2, y2] format. Your task is to generate "what" and "where" queries about objects not depicted in the image, based solely on its description. Respond to these queries by systematically confirming whether the queried object and its location are in the image, focusing on its absence from the image rather than its omission from the description. Keep the answers short. Please refer to the example below for the desired format.

[Description]: A man [147, 145, 238, 333] wearing blue overalls [143, 162, 235, 323] is standing next to a red minivan [107, 129, 496, 350] with an open door [195, 148, 298, 299].
[Query 1]: What is the woman wearing the blue overall standing next to? [Answer]: Find the woman. There is a man [147, 145, 238, 333] wearing a blue overall [143, 162, 235, 323] in the image, not a woman. The question is invalid.
[Query 2]: Where is the aeroplane? [Answer]: Find the aeroplane in the image. There is a red minivan [107, 129, 496, 350] in the image, not an aeroplane. This question is invalid.

[Description]: A person [112, 46, 244, 209] in a multicolored suit [115, 51, 240, 208] and helmet [197, 50, 240, 99] is performing a jump with a bike [74, 114, 286, 237] in a dull colored yard [[0, 90, 256, 331], [0, 230, 499, 331]] .
[Query 1]: What is the cat doing in the yard? [Answer]: Find the cat. There is a person [112, 46, 244, 209] in a multicolored suit [115, 51, 240, 208] performing a jump with a bike [74, 114, 286, 237] in a dull-colored yard [[0, 90, 256, 331], [0, 230, 499, 331]] in the image, not a cat. This question is invalid.
[Query 2]: Where is the dog playing? [Answer]: Find the dog in the image. There is a person [112, 46, 244, 209] in a multicolored suit [115, 51, 240, 208] performing a jump with a bike [74, 114, 286, 237] in a dull-colored yard [[0, 90, 256, 331], [0, 230, 499, 331]] in the image, not a tree. This question is invalid.

[Description]: <PLACE_HOLDER>

Figure 8: **Prompt utilized for creating negative samples in our SCouT dataset**, showcasing the method for generating queries that highlight contradictions or irrelevant information.

---

### In-context Object Extraction from Grounded Image Captions

[Task Description]: You are an intelligent bounding box annotator. I provide you with a caption of a photo that includes objects and their corresponding bounding box annotations. Your task is to extract objects which have corresponding bounding boxes next to them from the caption. Make a list, each of whose entries is a tuple, with the first item being the name of the object, and the second item being the bounding box coordinates corresponding to the object as (object name, [x1,y1,x2,y2]). Please refer to the example below for the desired format.

[Description]: A man [[24,36,76,97]] in a blue robe [[24,47,64,92]] is sitting on a white couch [[12,50,88,100]] with a cat [[33,58,50,70]] and a dog [[53,67,75,79]].
[Answer]: [("man", [24,36,76,97]), ("robe", [24,47,64,92]), ("couch", [12,50,88,100]), ("cat", [33,58,50,70]), ("dog", [53,67,75,79])]<stop>

[Description]: A group of people [[32,78,36,86; 40,77,43,86; 56,76,60,86; 46,77,50,87; 35,77,40,86; 50,77,54,86; 28,77,32,87]] are standing in a circle [[36,6,80,46]] watching kites [[36,6,80,46]] fly in the sky [[16,0,83,55]].
[Answer]: [("people", [32,78,36,86]), ("people", [40,77,43,86]), ("people", [56,76,60,86]), ("people", [46,77,50,87]), ("people", [35,77,40,86]), ("people", [50,77,54,86]), ("people", [28,77,32,87]), ("circle", [36,6,80,46]), ("kites", [36,6,80,46]), ("sky", [16,0,83,55])]<stop>

[Description]: <PLACE_HOLDER>

Figure 9: **Prompt used to extract object names and bounding boxes** from grounded image captions generated by our VLM models, illustrating the process of transforming free-form text outputs into structured data.

