# OpenReview forum: "Learning to Ground VLMs without Forgetting"
_ICLR.cc/2025/Conference — ICLR 2025 Conference Withdrawn Submission_

### Official Review · Reviewer_JPKR · 2024-11-02

**Soundness:** 3
**Presentation:** 2
**Contribution:** 2
**Rating:** 3
**Confidence:** 3

**Summary:**

The paper presents a method to extend a pretrained MoE VLM to do visual grounding tasks on images (Lynx).

The paper trains this method a new dataset and loss (collectively, SCouT). The dataset is based on Flickr30k images, with grounded  visual question answering labels, created by prompting a pretrained model (CogVLM) to do step-by-step grounded captioning.

Their specific model architecture (Lynx) adds a couple of experts to moe-llava, and dedicate those to visual grounding tasks using a learned very classifier to determine if the prompt is a grounding prompt. During finetuning, most of the weights remain frozen: pretrained visual encoder and multi-head attention weights. Then they train the experts as well as the classifier that learns to classify if this prompt is an image understanding or visual grounding prompt.

Lynx is trained on Referring Expressions (refCOCO and friends), and CogVLM pseudolabels for grounded image captioning on COCO and a new grounded VQA on top of Flickr30k (Scout).

They show results on 2D Object Det (subsets of COCO, LVIS, Pascal VOC), RefExp (RefCOCO/+/g). Because weights for image understanding tasks are largely frozen and routing is gated using MoE, the results on those tasks are essentially identical after training the additional additional -- which the authors show experimentally. They compare Lynx to the most similar method, PIN, as both methods involve finetuning pretrained models for grounding tasks, and show results on some standard object detection and referring expression benchmarks. There are some additional comparisons to CogVLM, Shikra, and others.

**Strengths:**

Many experiments on (mostly) standard datasets using (mostly) standard metrics makes it possible to guess the cross-reference with current literature.

**Significance:**
Visual grounding is an important problem, of great interest to the wider community.

**Originality:**
OK — add a few experts to existing architecture, create a new dataset using pseudolabels from existing models. Most of the innovation is on using these to create a “high-quality” pseudolabeled dataset using specific prompting strategies, and distilling the results into the adapted MoE (Lynx) model.


The distillation strategy freezes the weights used for non-grounding tasks, ensuring no performance regression as long as the classifier correctly detects this is not a grounding task.

The step loss uses something like teacher forcing, presumable to speed up training and make it more stable.

**Clarity:**
Good — the writing was pretty clear although there was some unnecessary detail/contributions.

**Quality:**
Good ablations on number of experts, amount of data used for Scout dataset. Some relevant external baselines are missing.

**Weaknesses:**

My main concern is about the experimental support for the claim in the conclusion, that "LynX demonstrates SOTA effectiveness in object localization on multiple visual grounding benchmarks and matches pretrained performance on image understanding benchmarks."

The paper presents comparisons for object localization on standard visual grounding datasets: PVOC, COCO, LVIS, RefCOCO, RefCOCO+, RefCOCOg. But some relevant baselines are not included in this paper, and the paper introduces new experimental protocols that makes the new numbers not comparable to SotA in other papers, directly. However, the baselines shown directly in this paper are not SotA for visual grounding benchmarks -- e.g. GroundingDino on COCO and LVIS object detection.

Both the main baseline in this paper (PIN) and the proposed method get around < 20 mAP on COCO in the most similar new protocol, while GroundingDino is close to SotA and gets > 60mAP  [2]. There is a similar gap for LVIS and the reported numbers also seem a bit lower than for RefCOCO/+/g, too.


### Baselines
These are a couple of baselines that would seem to be directly comparable to Lynx:

**Grounding Dino baseline:**

For example of something close to SotA for visual grounding, GroundingDino (ECCV 2024, https://github.com/IDEA-Research/GroundingDINO) is also an open-vocab grounding model that involves finetuning pretrained models for grounding tasks, and can ground multiple objects. It also reports numbers on the same grounding datasets (COCO/LVIS/PVOC/RefCOCO/+/G). Judging by GitHub stars, GroundingDino (6600 stars) is much more common than the main baseline PIN (24 stars) [1]. It can be pip installed and makes a pretty good baseline.

**MOE (MOE-LLAVA + Grounding DINO)**

Presumably the authors choose consider PIN to be the closest work because they are both "unified" end-to-end language generation models. Including comparisons against PIN is probably a useful comparison.

But if you are already freezing the weights of the some SotA model for image understanding (MoE-LLaVA), why not freeze weights for visual grounding (GroundingDino), too? Then Lynx just learns the mixture of experts classifier (the one that currently predicts $\alpha$,  depending on if the prompt is determined to be a visual grounding prompt).




[1] I definitely don't consider stars to be the main indicator of important, but it's a proxy :)
[2]  Perhaps I am misunderstanding which protocol is most comparable with COCO leaderboard: https://paperswithcode.com/sota/object-detection-on-coco. Protocol 3 seems most related to the standard leaderboard on COCO + LVIS

**Questions:**

- Scout: How does the performance of Scout change with different pseudoannotation models (e.g. alternatives to CogVLM)
- We also present an open-source evaluation pipeline for comparing VLMs at various granularities.
    - Can you clarify the motivation for this, since it seems to be one of the core contributions of the paper?
    - Seems like the protocol 3 is closest to what is in the current literature. Protocol 1 is supposed to be the most unconditional detection “list all objects”. And protocol 2 is an intermediary that asks a sentence transformer to paraphrase GT categories into more general ones, and use those for prompts.
    - The benefits for Protocols 1 + 2 is a bit unclear to me, since listing all objects unconditionally is very hard because getting the right semantic level for all prompts is basically impossible. This is why prompting is so useful! Anyway, adding protocols 1 + 2 doesn't seem to be necessary for using a model that produces text

---

### Official Review · Reviewer_aiHB · 2024-11-03

**Soundness:** 3
**Presentation:** 3
**Contribution:** 3
**Rating:** 5
**Confidence:** 3

**Summary:**

This paper proposes a method to fine-tune VLMs without catastrophic forgetting, especially in adapting VLMs for image understanding to grounding tasks. The authors leverage a dual-MoE architecture to pass the current task at hand to frozen and trainable expert heads. The resulting model, with the help of the synthesized chain-of-thought (SCouT) dataset to facilitate training for grounding, achieved significant improvement over previous state-of-the-arts in object grounding.

**Strengths:**

The proposed method is simple and intuitive, leveraging the MoE design for handling different tasks without catastrophic forgetting makes sense. The proposal of SCouT dataset is also beneficial for visual grounding tasks and learning the task classifier proposed in this paper.

**Weaknesses:**

One critical concern about the proposed model lies in its scalability. As also mentioned by the authors, scaling the number of experts significantly scales the parameters in VLMs. Under this assumption, the current fine-tuning without forgetting pipeline seems to strongly depend on the added new trainable expert heads. This raises several questions:

- To incorporate more tasks, can we smoothly get it done increasing the number of trainable branches? How to solve the scalability issue?

- As we can also do it in an iterative fashion, when adding new tasks to the current Lynx model, how should we do the job without significantly adding more parameters?

I guess the original purpose for fine-tuning was to have a model without too much additional overheads, though currently with one task the parameter doesn't increase, the scalability on many other tasks of the current pipeline seems to be a problem for me, so I hope the authors could address this concerns.

**Questions:**

See the weakness section.

---

### Official Review · Reviewer_hwLw · 2024-11-04

**Soundness:** 2
**Presentation:** 2
**Contribution:** 2
**Rating:** 3
**Confidence:** 3

**Summary:**

The main thesis of the paper is to incoriporate spatial output capabilities in VLMs without incurring catastorphic forgetting for image understanding tasks. The paper follows a prior work of PIN, but instead proposes to use additional mixture of experts (MoE) to tackle the forgetting — specifically it keeps the MoE’s for image understanding tasks frozen (and the attention weights frozen) and adds new trainable MoE for object recognition tasks. The paper also proposes a new dataset called SCoUT, which is a synthetic dataset built over Flickr Images for image recognition and object detection tasks. The results show that the proposed Lynx method does well on the object recognition tasks as well as the image recognition tasks.

**Strengths:**

The paper addresses a limitation of prior VLMS like PIN which forgets the image recognition tasks when finetuned for object recognition tasks. The methods section is clear and easy to follow. The obtained results are strong and back up the proposed contributions of this work

**Weaknesses:**

- The proposed strategy boils down to running the original VLM if the task is an image understanding task and the fine-tuned layers and heads if the task is object recognition. Unlike usual mixture of experts, where the selection of the experts is learned, the paper simply uses a pre-trained classifier to classify the task into image understanding or object recognition tasks, and based on that select the experts it uses. Put another way, I think the proposed strategy is to actually fine-tune a VLM for a new task (and let it forget the previously learned information) and simply choose whether to use the old weights (if it wants to tackle an old task) or new weights (if the task is new). As a methodology, I think this is trivial and not novel --- happy to be corrected on this if I am misunderstanding the paper.
- The experimental section is hard to follow — many sections do not explicitly link the experiment to the actual table number in the paper and it makes it hard to match the text to the proposed table.
- The experiments in protocol 1 and protocol 2 uses CogVLM as the ground-truth labels which is the same VLM used to generate the SCoUT dataset. Clealry the biases of CogVLM will come into play, and it is unclear if the benefits are becasue of scout data or just the biases. While protocol 3 does call out that the protocol 1 and 2 might not be perfect because “the GTs are limited by the quality of CogVLM”, I think it is more fair to clearly call out that the additional biases will creep in because the same VLM is used for generating data and evaluation.

**Questions:**

The main thesis of the paper, as mentioned in the paper is "Learning" to Ground VLMs without forgetting. However, the proposed pipeline seems to be a hard selection between two sets of weights: an old set of weights for image understanding tasks and the fine-tuned weights (which perhaps have forgotten about image understanding tasks) for object recognition. I would appreciate if the authors explain if I am misunderstanding the core premise of the paper and why the proposed solution is not a trivial usuage of different network weights.

---

### Official Review · Reviewer_4zcz · 2024-11-10

**Soundness:** 3
**Presentation:** 3
**Contribution:** 3
**Rating:** 3
**Confidence:** 4

**Summary:**

This paper introduces LynX, a framework for equipping pretrained Visual Language Models (VLMs) with visual grounding capabilities without forgetting their existing image and language understanding skills. It proposes a Dual Mixture of Experts (MoE) architecture that allows the model to specialize in both image understanding and visual grounding simultaneously. Additionally, it introduces SCouT (Synthetic Chain-of-Thought with Grounding), a high-quality synthetic dataset with step-by-step grounded chain-of-thought annotations to facilitate effective training on grounding tasks. The framework also includes a step-by-step training methodology that breaks down complex tasks into intermediate steps with individual loss functions.

**Strengths:**

(1)  LynX uses a dual MoE module with one frozen MoE pretrained on image understanding and another trainable MoE for new grounding capabilities.
     (2) The SCouT dataset provides rich supervision with step-by-step multimodal reasoning, an improvement upon previous synthetic datasets.
     (3) LynX outperforms larger models such as Shikra-7B on complex tasks like phrase grounding, despite having fewer parameters (1.67B vs 7B).

**Weaknesses:**

(1) The novelty of this paper is limited in its content. In summary, the key innovations of this paper can be regarded as CoT[1] combined with adapter[2, 3], which does not provide any new insights. Additionally, the step-by-step loss function is a common auto-regressive (next token prediction) loss. The BERT classification module is not an elegant solution for the method.
(2) The paper freezes the original network and only trains the added gated adapter. There is no doubt that the performance of the image understanding from the original base model cannot be affected. However, LynX, trained with additional data (REC, SCouT), shows limited performance improvement (83.9 -> 85.4) and is left behind by Shikra-7B.
Reference:
[1] Wei, J., Wang, X., Schuurmans, D., Bosma, M., Xia, F., Chi, E., Le, Q.V. and Zhou, D., 2022. Chain-of-thought prompting elicits reasoning in large language models. Advances in neural information processing systems, 35, pp.24824-24837.
[2] Zhang, L., Rao, A. and Agrawala, M., 2023. Adding conditional control to text-to-image diffusion models. In Proceedings of the IEEE/CVF International Conference on Computer Vision (pp. 3836-3847).
[3] Wang, W., Lv, Q., Yu, W., Hong, W., Qi, J., Wang, Y., Ji, J., Yang, Z., Zhao, L., Song, X. and Xu, J., 2023. Cogvlm: Visual expert for pretrained language models. arXiv preprint arXiv:2311.03079.

**Questions:**

The paper presents a straightforward solution to incorporate grounding ability into VLMs, but the innovation is not substantial enough to provide new insights. A paper that is to be accepted should offer the community fresh perspectives.

In Table 5, MoE-LLaVA-phi2 is the base model. It appears that Lynx has fewer active parameters, which may be a mistake. Comparing the MoE model with a non-MoE model is not fair in Table 5 and Table 4.

---

### Note · Authors · 2024-11-14

**Comment:**

We would like to thank the reviewers for their valuable feedback on the manuscript. Given the review scores, we would like to withdraw our submission.

**Withdrawal Confirmation:**

I have read and agree with the venue's withdrawal policy on behalf of myself and my co-authors.